# Genomic landscape of lymphatic malformations: a case series and response to the PI3Kα inhibitor alpelisib in an *N*-of-1 clinical trial

Montaser F Shaheen[1,2]*[†], Julie Y Tse[3][†], Ethan S Sokol[3], Margaret Masterson[4,5], Pranshu Bansal[6,7], Ian Rabinowitz[6,7], Christy A Tarleton[6,8], Andrey S Dobroff[6,8], Tracey L Smith[9,10], Thèrése J Bocklage[6,11], Brian K Mannakee[1,12], Ryan N Gutenkunst[1,13], Joyce Bischoff[14,15], Scott A Ness[6,8], Gregory M Riedlinger[4,16], Roman Groisberg[4,17], Renata Pasqualini[9,10], Shridar Ganesan[4,17]*, Wadih Arap[9,18]*

[1]University of Arizona Cancer Center, Tucson, United States; [2]Division of Hematology/ Oncology, Department of Medicine, University of Arizona College of Medicine, Tucson, United States; [3]Foundation Medicine, Inc, Cambridge, United States; [4]Rutgers Cancer Institute of New Jersey, New Brunswick, United States; [5]Department of Pediatrics, Rutgers Robert Wood Johnson Medical School, New Brunswick, United States; [6]University of New Mexico Comprehensive Cancer Center, Albuquerque, United States; [7]Division of Hematology/Oncology, Department of Internal Medicine, University of New Mexico School of Medicine, Albuquerque, United States; [8]Division of Molecular Medicine, Department of Internal Medicine, University of New Mexico School of Medicine, Albuquerque, United States; [9]Rutgers Cancer Institute of New Jersey, Newark, United States; [10]Division of Cancer Biology, Department of Radiation Oncology, Rutgers New Jersey Medical School, Newark, United States; [11]Department of Pathology, University of Kentucky College of Medicine and Markey Cancer Center, Lexington, United States; [12]Department of Epidemiology and Biostatistics, Mel and Enid Zuckerman College of Public Health, University of Arizona, Tucson, United States; [13]Department of Molecular and Cellular Biology, College of Science, University of Arizona, Tucson, United States; [14]Vascular Biology Program, Boston Children's Hospital, Boston, United States; [15]Department of Surgery, Harvard Medical School, Boston, United States; [16]Department of Pathology, Rutgers Robert Wood Johnson Medical School, New Brunswick, United States; [17]Division of Medical Oncology, Department of Medicine, Rutgers Robert Wood Johnson Medical School, New Brunswick, United States; [18]Division of Hematology/Oncology, Department of Medicine, Rutgers New Jersey Medical School, Newark, United States

**\*For correspondence:**
shaheenm@arizona.edu (MFS);
ganesash@cinj.rutgers.edu (SG);
wadih.arap@rutgers.edu (WA)

[†]These authors contributed equally to this work

## Abstract

**Background:** Lymphatic malformations (LMs) often pose treatment challenges due to a large size or a critical location that could lead to disfigurement, and there are no standardized treatment approaches for either refractory or unresectable cases.

**Methods:** We examined the genomic landscape of a patient cohort of LMs (*n* = 30 cases) that underwent comprehensive genomic profiling using a large-panel next-generation sequencing assay. Immunohistochemical analyses were completed in parallel.

**Results:** These LMs had low mutational burden with hotspot *PIK3CA* mutations (*n* = 20) and *NRAS* (*n* = 5) mutations being most frequent, and mutually exclusive. All LM cases with Kaposi sarcoma-like (kaposiform) histology had *NRAS* mutations. One index patient presented with subacute abdominal pain and was diagnosed with a large retroperitoneal LM harboring a somatic *PIK3CA* gain-of-function mutation (H1047R). The patient achieved a rapid and durable radiologic complete response, as defined in RECIST1.1, to the PI3Kα inhibitor alpelisib within the context of a personalized *N*-of-1 clinical trial (NCT03941782). In translational correlative studies, canonical PI3Kα pathway activation was confirmed by immunohistochemistry and human LM-derived lymphatic endothelial cells carrying an allele with an activating mutation at the same locus were sensitive to alpelisib treatment in vitro, which was demonstrated by a concentration-dependent drop in measurable impedance, an assessment of cell status.

**Conclusions:** Our findings establish that LM patients with conventional or kaposiform histology have distinct, yet targetable, driver mutations.

**Funding:** R.P. and W.A. are supported by awards from the Levy-Longenbaugh Fund. S.G. is supported by awards from the Hugs for Brady Foundation. This work has been funded in part by the NCI Cancer Center Support Grants (CCSG; P30) to the University of Arizona Cancer Center (CA023074), the University of New Mexico Comprehensive Cancer Center (CA118100), and the Rutgers Cancer Institute of New Jersey (CA072720). B.K.M. was supported by National Science Foundation via Graduate Research Fellowship DGE-1143953.

**Clinical trial number:**
 NCT03941782

## Editor's evaluation

The study examines the genomic landscape of a patient cohort of lymphatic malformations (LMs) through next-generation sequencing and immunocytochemistry. The authors identified actionable driver mutations in the P13KCA and NRAS genes. The study enhances our understanding of the genetic architecture of the otherwise disfiguring LMs in people.

## Introduction

Vascular anomalies, including lymphatic malformations (LMs), are usually diagnosed in children or young individuals and they can present as either isolated lesions or as part of somatic or congenital syndromes. Here, the term lymphatic malformation is used to include the clinicopathologic continuum of benign tumors of lymphatic origin (https://rarediseases.org/rare-diseases/lymphatic-malformations), including cystic lymphangioma, kaposiform lymphangiomatosis (KLM), and macro/microcystic LM. In general, LMs are managed by sclerotherapy, laser, or surgical interventions when there is an indication for therapy (*Perkins et al., 2010*). In certain cases, LMs can attain large sizes or involve critical locations, which poses treatment challenges such as the possibility of disfigurement. Genomic sequencing has demonstrated a somatic clonal origin for a number of nonmalignant growth conditions including LMs. Activating *PIK3CA* mutations have been reported in most pediatric patients with isolated or syndromic LMs (*Luks et al., 2015*). This finding has led to the use of mammalian target of rapamycin (mTOR) inhibitors for systemic therapy of unresectable LMs, given that mTOR is a molecule downstream of the PI3K pathway (*Fruman et al., 2017*). However, only a subset of patients responded, and the treatment can have substantial side effects. PI3K inhibitors have also been recently approved by the FDA for treatment of adults and children with severe manifestations *PIK3CA*-related Overgrowth Spectrum (termed PROS) who require systemic therapy, but the efficacy of alpelisib in isolated sporadic LMs is not at all clear. Activating *NRAS* mutations have been described in a subset of LM known as KLM (*Barclay et al., 2019*). KLM belong to a group of complex lymphatic anomalies that exhibit histologic and clinical features distinguishing them from classic LM. It is not as yet clear which oncogenic drivers, if any, are present in LMs with wild-type *PIK3CA* and *NRAS* alleles.

To define the spectrum of genomic alterations and lesions present in LMs, here we have analyzed a patient cohort of LMs (*n* = 30 cases) assayed by clinical-grade genomic sequencing. Pathogenic activating mutations in *PIK3CA* and *NRAS* were the most common genetic alterations found. Strikingly, the *PIK3CA* and *NRAS* mutations were mutually exclusive with *NRAS* mutations being greatly

enriched in LMs with kaposiform morphology. We have also performed an *N*-of-1 trial of the PI3Kα inhibitor alpelisib in a young man with an activating *PIK3CA* point mutation, presenting with a giant (unresectable) retroperitoneal and pancreatic LM, who had a dramatic and prolonged response to the drug lasting years, and we present confirmatory translational correlates in vitro.

## Materials and methods

### Genomics and DNA sequencing

Hybrid-capture DNA sequencing targeting exons of at least 324 cancer genes and select introns of 36 genes were performed on the patient samples; a subset ($n = 2$) were also analyzed with plus RNA sequencing of 265 genes to improve rearrangement detection. A total of 30 patient samples were sequenced with either the DNA-only assay ($n = 28$; Foundation One CDx, Foundation Medicine; Cambridge, MA) or the DNA + RNA assay ($n = 2$; Foundation One Heme, Foundation Medicine; Cambridge, MA).

### Immunohistochemistry

Immunohistochemistry (IHC) was performed on formalin-fixed, deparaffinized, 5-µm-thick sections mounted on charged slides. Antibodies to P-AKT (Ser473) and P-6S (Ser240/Ser244) were obtained from Cell Signaling Technology, Danvers, MA. Diaminobenzidine was used as the chromogen and hematoxylin as the counterstain. All stages of staining were carried out on an automated system (Ventana Discovery Research Instrument; Ventana, Tucson, Arizona). Positive and negative controls were appropriately reactive. A surgical pathologist with subspecialty interest in musculoskeletal pathology (T.J.B.) interpreted the results.

### Lymphatic malformation-lymphatic endothelial cell sensitivity to alpelisib in vitro

Lymphatic malformation-lymphatic endothelial cells (LM-LECs) were maintained as described (*Boscolo et al., 2015*) and negative for mycoplasma at the time of these studies. Mycoplasma test was performed using the MycoAlert Mycoplasma Detection Kit (Cat # LT07-218, Lonza) following the manufacturer's instructions. Real-time analysis of cell viability was performed by using the xCELLigence system RTCA SP (ACEA Biosciences). Briefly, $5 \times 10^3$ LM-LECs per well were seeded in an E-Plate 96 (ACEA Biosciences) and cell proliferation was recorded hourly. When the cells reached the exponential growth phase, new media containing alpelisib at 1, 3, 10, 30, or 100 nM was added and alpelisib cytotoxic effect was recorded hourly. $IC_{50}$ was calculated by using the dose–response curve function available in the xCELLigence software Version 2.0. Cell index (%) reflects cell viability.

### Clonogenic survival assays

For the clonogenic survival assay, the LM-LECs were trypsinized, counted, and plated in complete growth media on 6-well plates (Falcon) (400 cells/well). Seven days later, alpelisib (at the empirically determined $IC_{50}$ from a standard calibration curve) was added in duplicate wells. After 24 or 48 hr of incubation, cells were fixed and stained in 50% methanol in water containing 0.3% crystal violet to facilitate counting of colonies (≥50cells).

### Statistics

All values are expressed as mean with error bars expressed as standard deviation. For comparison between untreated (negative), dimethyl sulfoxide control, and alpelisib-treated LM-LEC cells, the ordinary one-way analysis of variance and Tukey's multiple comparisons test with a single pooled variance were used. Statistical analysis was performed using the GraphPad Prism 7.0d software (GraphPad Software Inc, San Diego, CA). Fisher's exact test was used for categorical data, owing to the sizes of the cohorts. A two-tailed p value of $<0.05$ was considered to be statistically significant.

### Study approval

Approval for this study, including a waiver of informed consent and Health Insurance Portability and Accountability Act waiver of authorization, was obtained from the Western Institutional Review Board (IRB; protocol #20152817). A single-institution personalized clinical protocol to treat the patient with

the experimental PI3Kα inhibitor alpelisib was scientifically reviewed by the Protocol Review and Monitoring Committee (PRMC) and approved by the local Institutional Review Board (IRB) of the University of New Mexico Comprehensive Cancer Center. The study (NCT03941782) was conducted in accordance with the protocol, Good Clinical Practice guidelines, and the provisions of the Declaration of Helsinki. CARE reporting guidelines were also used for this patient (*Gagnier et al., 2013*). The index patient signed an informed written consent form.

## Results

### Mutational landscape and histopathology of LMs

A set of 30 cases of LMs (from 30 individual patients) were assayed with genomic profiling at Foundation Medicine, Inc (Cambridge, MA). Twenty-eight cases were sequenced using hybrid-capture next-generation sequencing (NGS) targeting exons of 300 + cancer genes and select introns of 36 genes. Two other cases were sequenced using hybrid-capture based DNA sequencing targeting exons of 406 + cancer genes and select introns of 36 genes, plus RNA sequencing of 265 genes for rearrangement calling. The patients were predominantly pediatric age (median 9-year-old; range, 1- to 45-year-old), with a slight female predominance (17 females, 57%–13 males, 43%). Seven patients had a documented history of prior treatment with an mTOR inhibitor, such as sirolimus. Seven patients (23%) had documentation of clinical diagnoses of overgrowth syndromes including Congenital Lipomatous Overgrowth with Vascular, Epidermal, and Skeletal anomalies (termed CLOVES), Klippel–Trenaunay syndrome, and phosphatase and tensin homolog (PTEN)-like hamartoma syndrome at the time of testing. Twelve patients (40%) had multifocal disease and eight patients had involvement of bone and visceral sites (*Table 1*). Expert histopathological review showed that only four (13%) had kaposiform morphology, while 26 (87%) had conventional histology. The estimated histopathologic purity ranged from 10% to 70% (median 20%).

Mutational profiling showed that these LMs had uniformly low tumor mutational burden (median, zero mutations/Mb; range, 0–2.6 mutations/Mb), and none had evidence of microsatellite instability. The most common mutations were activating mutations in *PIK3CA*, seen in 20 (67%), and activating *NRAS* mutations, seen in 5 (17%) (*Figure 1A, B*). The *PIK3CA* mutations included hotspot mutations in both the helical domain and the kinase domain (*Samuels et al., 2004*). The *NRAS* mutations all altered the known hotspot at residue glutamine 61 (Q61) in the phosphorylation binding loop. Of the five patients (17%) with no alterations in *PIK3CA* and *NRAS*, one case (Patient #29; *Table 1*) had an activating *GOPC–ROS1* fusion (*Figure 1C*) with a *ROS1* missense point mutation. Similar *GOPC–ROS1* fusions have been reported in pediatric gliomas in the setting of microdeletion of chromosome 6q22[8], and have also been found in adult lung cancer (*Drilon et al., 2021*).

The variant allele frequencies (VAFs) of the *PIK3CA* and *NRAS* mutations were relatively low (median, 6%; range, 1–38%), compatible with relatively low histopathologic estimated percentage of tumor nuclei (%TN) to overall cellular nuclei (median, 20%; range, 10–70%). These results suggest that the *PIK3CA* and *NRAS* mutations were likely clonal, but in the setting of relatively low tumor purity in the specimens.

### Enrichment of *NRAS* mutations in LMs with kaposiform features

Histopathological analysis of the lesions by an board-certified dermatopathologist (J.Y.T.) identified that four (13%) of the analyzed specimens had kaposiform histopathological features with highly cellular, clustered, or sheet-like, proliferation of spindled lymphatic cells admixed with dilated thin-walled lymphatic vessels (*Figure 1D*). The remaining 26 lesions (87%) had conventional histopathological features of classic LM, with proliferation of dilated, thin-walled lymphatic vessels with or without luminal proteinaceous material. Lymphatic phenotype of the cells was confirmed by immunopositivity for PROX1 or D2-40, by report. Of the conventional histology LM cases (*n* = 26), 20 (77%) had a *PIK3CA* mutation, while 1 (4%) had a *NRAS* mutation, and 5 (19%) were wild-type for both genes, including a single case with a *GOPC–ROS1* genetic fusion. Notably, all four cases of LM with kaposiform features had an activating *NRAS* mutation, consistent with enrichment of *NRAS* mutation (p = 0.00018) and lack of *PIK3CA* mutation in this histology (p = 0.0046). The lone *NRAS*-mutant LM with conventional histology was a small core needle-biopsy specimen of a large visceral tumor, raising the possibility that the histopathologic features of the sampled tissue may not have been representative

**Table 1.** Clinical and histological features of lymphatic malformation cohort.

| Patient | Age (years) | Sex | Submitted clinical syndrome | Localized vs. multifocal | Location of LM(s) | Specimen type | LM histology | PIK3CA or NRAS alteration | % VAF |
|---|---|---|---|---|---|---|---|---|---|
| 1 | 9 | M | CLOVES | Multifocal | Superficial soft tissues | Excision | Conventional | PIK3CA E542K | 14 |
| 2 | 4 | F | | Localized | Superficial soft tissues | Excision | Conventional | PIK3CA E542K | 7 |
| 3 | 1 | F | | Localized | Superficial soft tissues | Excision | Conventional | PIK3CA H1047R | 11 |
| 4 | 17 | M | | Localized | Superficial soft tissues | Excision | Conventional | PIK3CA H1047R | 4 |
| 5 | 18 | M | | Localized | Superficial soft tissues | Excision | Conventional | PIK3CA H1047L | 4 |
| 6 | 8 | F | Klippel–Trenaunay | Localized | Superficial soft tissues | Excision | Conventional | PIK3CA H1047R | 9 |
| 7 | 9 | M | | Localized | Visceral | Core biopsy | Conventional | PIK3CA E545K | 7 |
| 8 | 3 | F | | Localized | Superficial soft tissues | Excision | Conventional | PIK3CA C420R | 5 |
| 9 | 23 | M | | Localized | Visceral | Incisional biopsy | Conventional | PIK3CA H1047R | 4 |
| 10 | 16 | F | PTEN-like hamartoma | Localized | Superficial soft tissues | Excision | Conventional | PIK3CA H1047R | 3 |
| 11 | 3 | F | CLOVES | Multifocal | Superficial soft tissues | Excision | Conventional | PIK3CA E545K | 12 |
| 12 | 1 | M | | Multifocal | Superficial soft tissues | Excision | Conventional | PIK3CA H1047R | 2 |
| 13 | 4 | F | | Localized | Superficial soft tissues | Excision | Conventional | PIK3CA E542K | 6 |
| 14 | 5 | M | | Localized | Superficial soft tissues | Excision | Conventional | PIK3CA H1047R | 5 |
| 15 | 1 | F | | Localized | Superficial soft tissues | Excision | Conventional | PIK3CA E545K | 1 |
| 16 | 14 | F | | Multifocal | Visceral | Excision | Conventional | PIK3CA C420R | 14 |
| 17 | 2 | F | CLOVES | Multifocal | Superficial soft tissues | Excision | Conventional | PIK3CA C420R | 38 |
| 18 | 16 | F | CLOVES | Localized | Superficial soft tissues | Excision | Conventional | PIK3CA E453K | 32 |
| 19 | 10 | F | CLOVES | Multifocal | Superficial soft tissues | Excision | Conventional | PIK3CA H1047L | 15 |
| 20 | 9 | M | | Localized | Superficial soft tissues | Excision | Conventional | PIK3CA H1047R | 5 |
| 21 | 9 | F | | Multifocal | Visceral | Excision | Kaposiform | NRAS Q61R | 5 |
| 22 | 8 | M | | Multifocal | Superficial soft tissues | Excision | Kaposiform | NRAS Q61R | 5 |
| 23 | 9 | F | | Multifocal | Visceral | Excision | Kaposiform | NRAS Q61R | 1 |
| 24 | 45 | M | | Multifocal | Visceral | Core biopsy | Conventional | NRAS Q61R | 6 |
| 25 | 10 | F | | Localized | Superficial soft tissues | Core biopsy | Kaposiform | NRAS Q61R | 14 |
| 26 | 17 | M | | Multifocal | Superficial soft tissues | Excision | Conventional | WT | NA |
| 27 | 24 | M | | Localized | Bone | Core biopsy | Conventional | WT | NA |
| 28 | 3 | M | | Multifocal | Superficial soft tissues | Excision | Conventional | WT | NA |
| 29 | 11 | F | | Localized | Superficial soft tissues | Excision | Conventional | WT | NA |
| 30 | 9 | F | | Localized | Superficial soft tissues, bone | Biopsy | Conventional | WT | NA |

CLOVES – congenital lipomatous overgrowth, vascular anomalies, epidermal nevi, and skeletal anomalies; NA – not applicable; VAF – variant allele frequency of PIK3CA or NRAS.

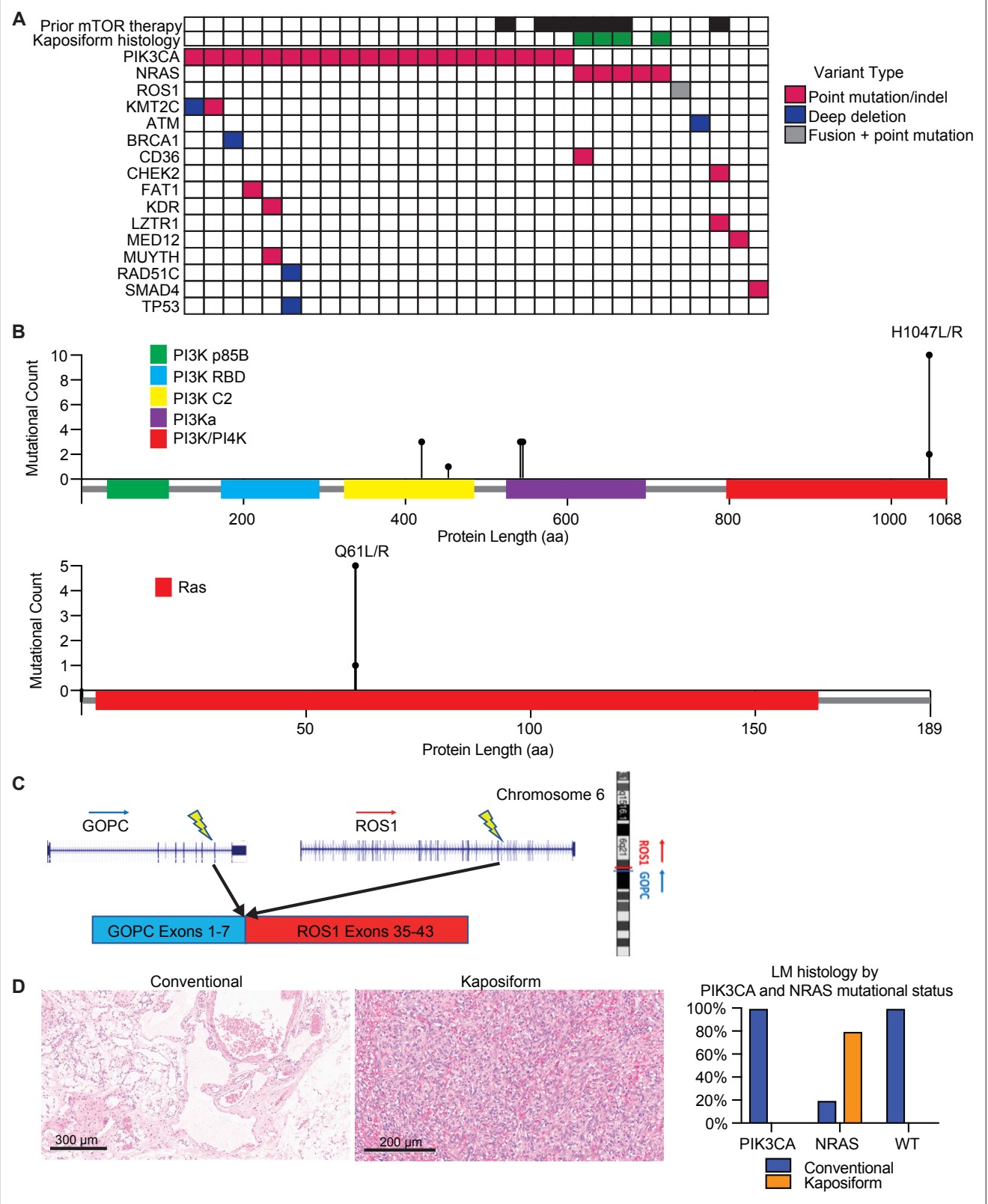

**Figure 1.** Mutational landscape and histopathology of lymphatic malformations (LMs). (**A**) Oncoprint showing mutational landscape of 30 LM samples sequenced. (**B**) Lollipop plot showing spectrum of *PIK3CA* and *NRAS* mutations in this cohort. (**C**) Schema showing details of *GOPC–ROS1* fusion identified in an *NRAS* and *PIK3CA* wild-type LM. (**D**) Representative histologic images for LMs with conventional and kaposiform histology. The relative frequencies of *PIK3CA* and *NRAS* mutations in the two histologic variants are plotted.

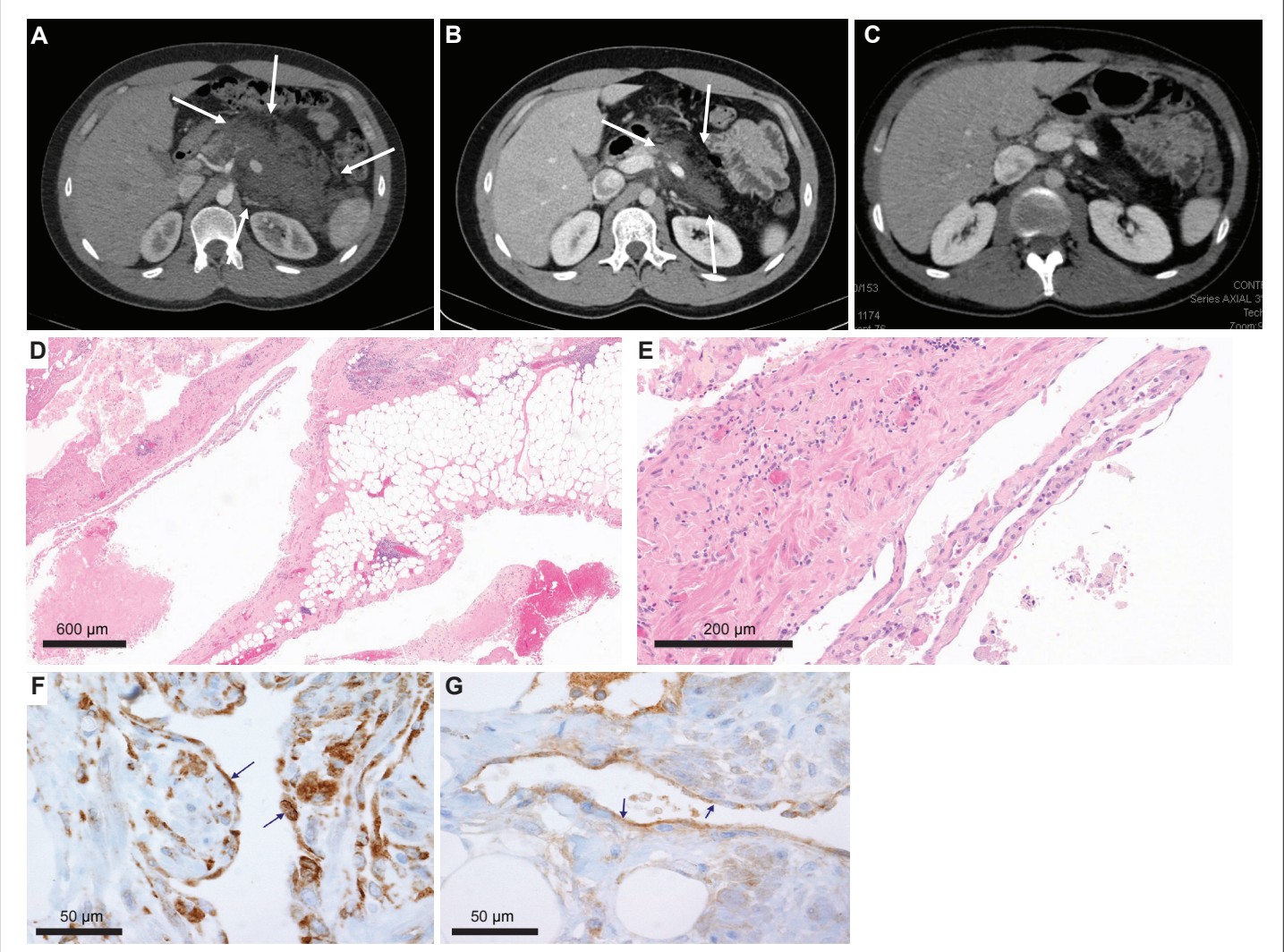

**Figure 2.** Imaging and histological analysis of lymphatic malformation (LM) patient. (**A**) Baseline CT abdomen scan at the time of presentation demonstrating a large retroperitoneal/pancreatic LM. (**B**) CT abdomen scan 6 weeks after the initiation of alpelisib. (**C**) CT abdomen scan 1 year into the trial. (**D, E**) Hematoxylin and eosin (H&E)-stained photomicrographs of the LM showing dilated lymphatic channels percolating through visceral fat and associated patchy lymphocytic inflammation (×4 and ×20, respectively). (**F**) Immunohistochemistry utilizing an anti-P-6S antibody demonstrates PI3Ka pathway activation within the channels' lining cells. (**G**) Anti-P-AKT positivity in the lining endothelium of lymphatic channels as well.

of the entire lesion due to the histologic spatial heterogeneity often seen in LMs with kaposiform histology. Additional histopathologic features were assessed, including altered adipose tissue, muscularized blood vessels, vascular endothelial cell atypia, and inflammation; no statistical significance was identified between the four *NRAS*-mutant LM cases and the remainder of the patient cohort.

## Case report and *N*-of-1 clinical trial results

One of the conventional histology LMs was a 23-year-old male with no significant medical or family history who presented with subacute abdominal pain (Patient #9, *Table 1*). He was hospitalized and his exam revealed a distended abdomen that was tender to palpation. A computed tomography exam revealed a large solid mass based on the retroperitoneal area and the pancreas (*Figure 2A*), and a neoplastic process was suspected. A core needle biopsy was attempted but yielded no definitive tissue diagnosis. An open laparoscopic surgical biopsy was performed and revealed a vascular tumor with features of a giant retroperitoneal and pancreatic LM (*Figure 2D, E*). After discussing a surgical approach, the patient and the surgical team decided not to proceed due to the complexity

of surgical resection and associated risks. The tissue was submitted for NGS to identify potential biomarkers for targeted therapy.

Clinical-grade sequencing of the biopsy sample from Patient #9 uncovered a single activating point mutation in *PIK3CA* (H1047R). All other genes in the panel were wild-type except for another unit of the PI3K complex (*PIK3C2B*) that showed a variant (R458Q) of unknown significance (VUS). To confirm activation of the PI3Kα pathway, we performed IHC staining of the downstream targets (P-AKT and P-6S), and, as predicted, these phosphorylation events were detected in the lining cells of the abnormal lymphatic channels (*Figure 2F, G*).

Based on the genomic profile, we designed and offered this young man a single-patient (*N*-of-1) personalized clinical trial of the PI3Kα inhibitor alpelisib (NCT03941782), which at the time was still investigational (non-FDA approved). Screening procedures included an echocardiogram that revealed an ejection fraction (EF) of 47%. A cardiac MRI confirmed a low EF with no infiltrative process or other abnormalities. Paradoxically, the patient was completely asymptomatic from a cardiac standpoint and he was able to run two miles on a daily basis. We hypothesized that the decreased EF, in the absence of accompanying clinical signs or symptoms of heart failure, was likely artefactual due to hemodynamic changes related to the very large circulatory volume sequestration in his abdomen.

The patient was started on alpelisib daily dose of 350 mg orally (*Juric et al., 2018*) and he reported regression of his abdominal bulge within a few days. He reported no adverse events and was closely monitored for hyperglycemia. Repeated echocardiogram 2 months later showed normalization of the EF. A CT scan of the abdomen done 6 weeks into the trial revealed remarkable shrinkage of the LM (*Figure 2B*). Follow-up CT scans showed progressive reduction until complete response at 1 year of trial initiation (*Figure 2C*). The patient continued to do well on maintenance alpelisib for 2 years with no evidence of progression. After 2 years, alpelisib was discontinued due to theoretical concerns about long-term adverse impact on vascular homeostasis. Unfortunately, the mass recurred after a few weeks so the patient was resumed on alpelisib with a second deep partial response, which is still ongoing for over 3 years.

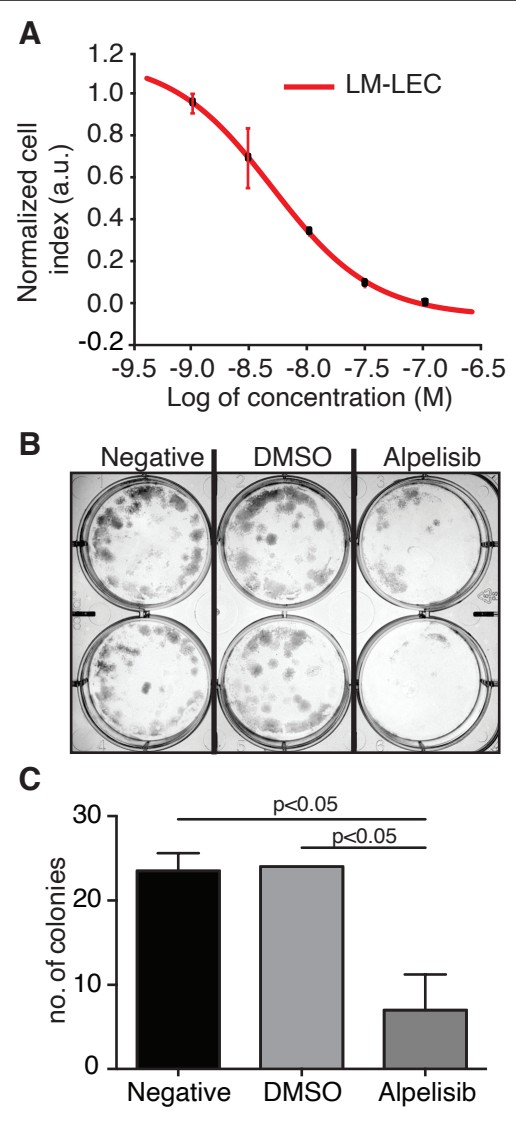

**Figure 3.** Alpelisib reduces lymphatic malformation-lymphatic endothelial cell (LM-LEC) viability. (**A**) Logarithmic dose–response curve of alpelisib was performed using the xCELLigence RTCA system. 1, 3, 10, 30, and 100 nM (*n* = 5 replicates) of alpelisib were used to determine the concentration–response curve. The alpelisib half maximal inhibitory concentration (IC$_{50}$) was calculated for LM-LEC at 24 hr after treatment as $4.72 \times 10^{-9}$ M. Error bars are shown as mean +/-standard deviation (SD), which was automatically calculated for each data point by the xCELLigence RTCA system software (Version 2.0) based on five replicates per drug concentration. (**B**) Illustrative picture of LM-LEC clonogenic plaques at 24 hr after alpelisib treatment ($4.72 \times 10^{-9}$ M). Negative, no treatment; dimethyl sulfoxide (DMSO), vehicle control. Experiments were performed two times with similar results. LM-LEC colonies were stained with crystal violet (0.3%). (**C**) Colony count 24 hr after alpelisib treatment ($4.72 \times 10^{-9}$ M; *n* = 2 wells/condition). Error bars are

*Figure 3 continued on next page*

*Figure 3 continued*

shown as mean +/- SD calculated by GraphPad Prism by determining the square root of variance for each data point deviation relative to the mean.

## Alpelisib inhibits primary PI3Kα-mutant LM-derived endothelial cells

We have also investigated the concentration-dependent effects of alpelisib on LM-LECs isolated from a surgically resected specimen (*Boscolo et al., 2015*). Targeted sequencing of DNA from LM-LECs identified a somatic missense mutation in *PIK3CA* (H1047L), the same locus altered in our alpelisib-treated patient and the site of half of the *PIK3CA* alterations in the LM cohort studied (*Table 1*). In addition, a nonsense mutation of the regulatory PI3K unit *PIK3R3* (R309*) was also detected in the CD31-positive LM-LECs and CD31-negative nonendothelial cells isolated from the same LM, indicating its germline origin (*Boscolo et al., 2015*). We investigated the effect of alpelisib on the growth of LM-LECs and a concentration-dependent response curve was observed (*Figure 3*). The IC$_{50}$ of alpelisib against LM-LECs was empirically determined in vitro to be $4.72 \times 10^{-9}$ M at 24 hr. This in vitro translational model confirms the sensitivity of LM-derived human cells containing a target H1047$^{R}/_{L}$ mutation to alpelisib.

## Refined genomic and sequencing analyses

We performed whole-genome sequencing (WGS) on paired LM/germline DNA from our index patient to explore the mutational profile beyond the genes that were probed in the Clinical Laboratory Improvement Amendments (CLIA)-approved clinical sequencing assay. The *PIK3CA* H1047R mutation was identified with a VAF of 11%. This finding is consistent with the ≤10% rate of mutant cells, and low tumor cellularity of LMs with *PIK3CA* mutation (*Luks et al., 2015*). Few other somatic coding mutations were identified in the LM tissue (*Supplementary file 1*).

To gain further molecular mechanistic insight, we have also performed RNA-seq studies to identify gene expression patterns within the LM sample from our index patient compared to normal tissue (*Figure 4*). RNA-seq data of biopsy samples from Patient #9 (*n* = 2 samples; *Figure 4A*, Group A) were compared to several normal human control tissue samples from bladder, colon, kidney, and salivary gland (*n* = 4, one sample per each tissue; *Figure 4A*, Group B). There is little difference between the two LM samples, but, by using an arbitrary cutoff of at least twofold up or down with adjusted p values of 0.05 or less, we identified 668 upregulated and 850 downregulated genes. The heatmap summarizes the results of the differential gene expression analysis; 125 genes are shown. The volcano plot summarizes the distribution of genes that were differentially expressed (*Figure 4B*). Here, the vertical axis shows the p value and the horizontal axis shows the fold-change. The genes that were more than twofold changed and had an adjusted p value less than 0.05 are shaded red. Similar numbers of genes were up- and downregulated. Several of the most highly induced genes, *CHI3L1*, *GPX1*, *PLIN1*, *PLIN4*, and *JAK3*, have been linked to enhanced growth or cell survival in other tumor types (*Cheng et al., 2019*; *Qiu et al., 2018*; *Sirois et al., 2019*; *Vadivel et al., 2021*; *Zhang et al., 2020*). Finally, a preliminary Gene Ontogeny (GO) analysis (*Subramanian et al., 2007*) of Patient #9 LM revealed enrichment of mRNA of genes involved in vascular development, cell motility, inflammatory response, positive regulation of response to stimuli, blood vessel morphogenesis, among others; notably, the kinase *JAK3* gene was one of the highest expression mRNAs in the LMs compared to normal tissue controls.

## Discussion

Here, we report the mutational landscape of a patient cohort of LMs (*n* = 30 cases) which underwent comprehensive genomic profiling. We have confirmed prior reports that hotspot activating mutations in *PIK3CA* are common driver events in these lesions, seen in 20 (67%) of these cases. Interestingly, *NRAS* mutations were seen in an additional five (17%) cases and were particularly enriched in LMs with a kaposiform histopathology. This finding supports previous studies that have shown that LMs with kaposiform features likely represent a distinct entity (KLA) (*Barclay et al., 2019*; *Croteau et al., 2014*). KLM have distinct clinical, histologic, and genomic features. Clinically, they are more likely to occur in young patients and commonly present as generalized processes with involvement of the mediastinum, pleura, and pericardium. Histologically, they are composed of highly cellular sheet-like and nodular proliferations of spindle cells, reminiscent of Kaposi sarcoma (*Croteau et al., 2014*). Unlike Kaposi sarcoma, the tumor cells lack immunopositivity for human herpesvirus-8 (HHV-8) latency-associated

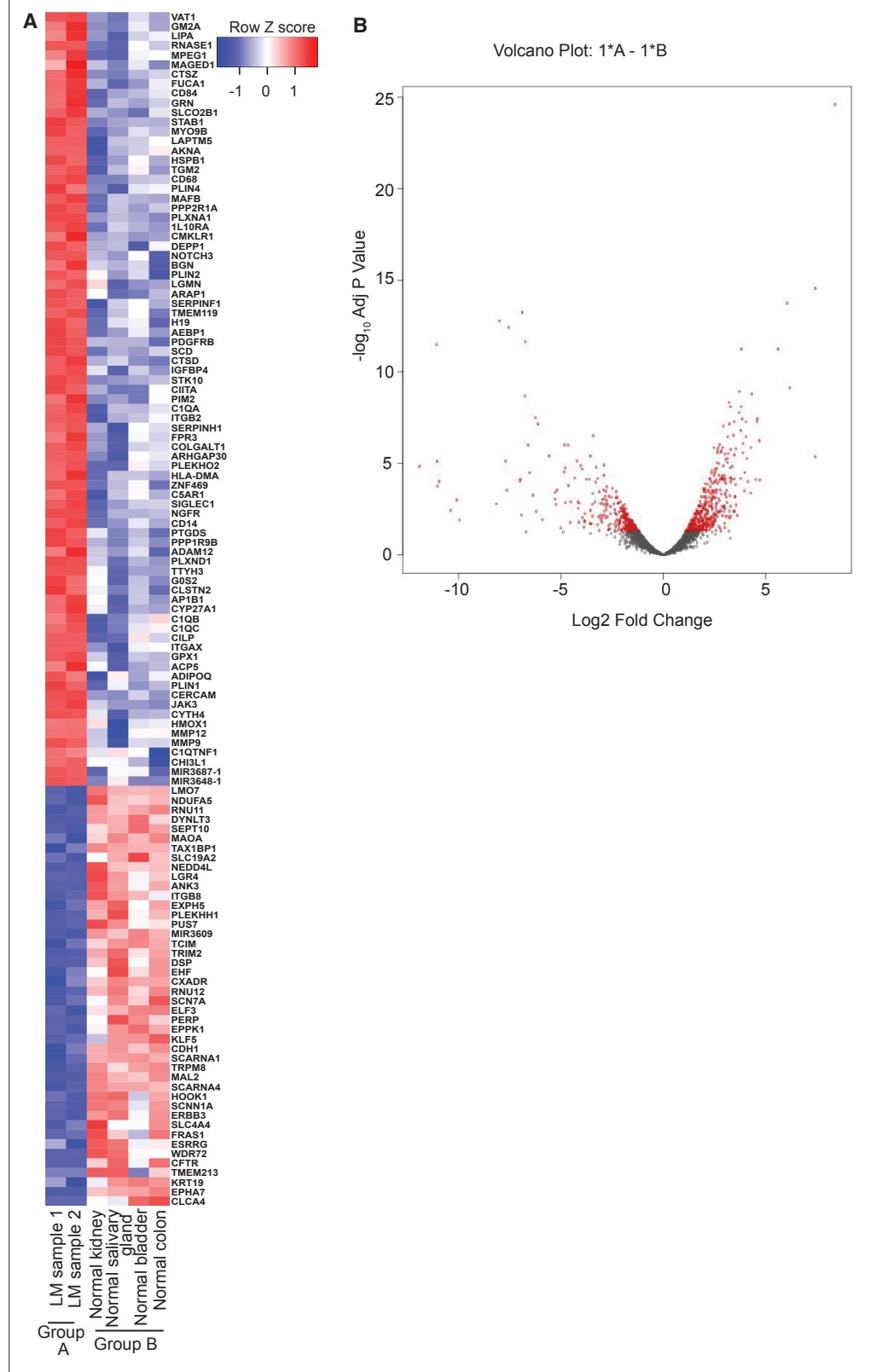

**Figure 4.** RNA-seq analysis of lymphatic malformation (LM) samples from index patient (#9). (**A**) The heatmap summarizes the results of the differential gene expression analysis. Up- and downregulated genes are shaded red and blue, respectively. (**B**) The volcano plot summarizes the distribution of genes that were differentially expressed. The vertical axis shows the p value and the horizontal shows the fold-change. The genes that were more than

*Figure 4 continued on next page*

*Figure 4 continued*

twofold changed and had an adjusted p value less than 0.05 are shaded red. Similar numbers of genes were up- or downregulated.

nuclear antigen (LANA). Genomically, recent studies have shown they tend to harbor somatic activating alterations in *NRAS* (*Barclay et al., 2019Barclay et al., 2019*). As a caveat, for the one *NRAS*-mutant LM with classic histology, the histologic classification was based on a small biopsy, and it is certainly possible that kaposiform histology was present in the large visceral LM but not captured by the limited sampling by core needle biopsy. Importantly, three of the five patients (60%) with *NRAS*-mutant LMs had failed treatment with sirolimus prior to NGS. There are reports that some *NRAS*-mutant LMs may respond to treatment with MEK inhibitors (*Dummer et al., 2017*), suggesting this may be an option for LMs with kaposiform features.

Of the five cases without either *PIK3CA* or *NRAS* mutations, all of classic histology, a single case had a known pathogenic in-frame *GOPC–ROS1* genetic fusion predicted to have an intact ROS1 kinase domain and thus potentially function as the driver. Similar *GOPC–ROS1* fusions have been seen in pediatric gliomas and adult lung cancers and may be sensitive to ROS1 inhibitors (*Davare et al., 2018*; *Drilon et al., 2021*). These data suggest that most LMs may have a potentially actionable driver mutation, with *PIK3CA* mutations dominating LMs with conventional histology and *NRAS* mutations predominantly or exclusively seen in the minor subset of LMs with kaposiform features. It is possible that the other *NRAS* and *KRAS* wild-type LMs may also have oncogenic alterations in other members of the *PIK3CA* or MAPK signaling pathway members that were not profiled by targeted sequencing strategies. We appreciate that one limitation of our study is that the cohort presented in *Table 1* was established from clinical information provided by the ordering physicians early in the course of diagnostic investigation. Therefore, we cannot rule out that the working clinical diagnosis and/or pathologic diagnoses were refined after genomic analyses without transmission of these data to the reference laboratory that supplied the cohort data. Comprehensive NGS analysis of LMs with *PIK3CA* and *NRAS* wild-type may be required to identify any potential actionable driver mutations. In patients without solid LM tissue available for NGS, liquid biopsy—or NGS performed on circulating tumor DNA (ctDNA) in peripheral blood—may be a possible solution for LMs, which are innately associated with the vascular system and thus potentially 'shedding' ctDNA into the peripheral blood.

To illustrate the potential for therapeutic intervention of the target mutations identified, we performed an *N*-of-1 trial of alpelisib in one young adult index patient with a giant retroperitoneal and pancreatic LM with conventional histological features and a gain-of-function H1047R somatic *PIK3CA* mutation. Our index patient experienced a rapid, complete, and durable clinical response with this small molecule PI3Kα inhibitor. Given the high frequency of *PIK3CA* mutations in pediatric LMS (*Luks et al., 2015*), this finding suggests that alpelisib may be highly effective for systemic, nonsurgical treatment approach to this class of disorders. Furthermore, the lack of toxicity to alpelisib in our case is promising in terms of a potential future treatment of young patients with LMs. Our patient did not experience increases in glucose levels, consistent with reported lack of alpelisib-induced hyperglycemia in most pediatric patients with PROS (*Venot et al., 2018*; *Mayer et al., 2017*). In this prior series, only one patient developed new-onset hyperglycemia and this was controlled by dietary modification (*Venot et al., 2018*). These findings suggest that the effect of alpelisib on inducing hyperglycemia might perhaps be less of a concern in younger patients, who may have more robust glucose homeostasis, compared with older patients who may already have subclinical insulin resistance.

Ultimately, we decided to hold alpelisib after 2 years of complete radiological response, and unfortunately the LM relapsed but the patient still achieved a major partial response on the second challenge with alpelisib. This result suggests that PI3Kα inhibitors do not completely eradicate all LM-initiating cells, and they may need to be given long term (in our young index patient case, perhaps over decades) in *PIK3CA*-mutant LMs for sustained control. This class of drugs can also be envisioned to be utilized in a neoaduvant approach to render large cases resectable. Our patient declined surgery after initial response and he continues on alpelisib for several years. Acquired resistance mechanisms to PI3Kα inhibitors have been reported, due to other associated compensatory or bypassing mutations such as ones involving *RAS* oncogene (*Janku et al., 2014*) or *PTEN* tumor suppressor gene (*Juric et al., 2017*), and these may conceivably arise in these patients with longer follow-up over time. Deftly balancing the potential benefits of continuing treatment with the potential for drug resistance

mechanisms will require monitoring for both actionable known and novel mutations through NGS of LM tissue samples or liquid biopsy.

In a series of pediatric patients with LMs, Luks et al. identified *PIK3CA* gene mutations in patients with sporadic LMs in 16 out of 17 patients (94%) or syndromic LMs such as the Klippel–Trenaunay syndrome in 19 out of 21 patients (90%), fibro-adipose vascular anomaly in 5 out of 8 patients (63%), along with the CLOVES syndrome in 31 out of 33 patients (94%) (*Luks et al., 2015*). H1047R was one of the top 2 most frequently encountered hotspot mutations in this series. Venot et al. reported a single arm clinical trial of alpelisib in 19 patients with pediatric PROS including CLOVES (*Venot et al., 2018*). Alpelisib treatment-induced clinical responses in all patients, including improvement of cardiac EF as seen in our index patient. Of note, alpelisib-induced responses in patients who did not respond to prior treatment with mTOR inhibitors, such as rapamycin, similar to observations in *KRAS*-mutant oncology patients (*Di Nicolantonio et al., 2010*), similar to the recent findings of Delestre et al. (*Delestre et al., 2021*). Small clinical series have shown that mTOR inhibition can induce responses in a subset of unselected advanced LMs, with observed response rates of ~50–60% (*Freixo et al., 2020*). The on driver-oncoprotein activity, higher response rates, and tolerability suggest alpelisib may be more effective than mTOR inhibitors in this setting. It is tempting to speculate that a wide variety of *PIK3CA*-mutant somatic overgrowth conditions (*Hucthagowder et al., 2017*) may be amenable to medical treatment with FDA-approved PI3Kα inhibitors, either as neoadjuvant treatment in potentially resectable cases, or as primary treatment in unresectable cases (*Juric et al., 2018*; *Juric et al., 2017*; *Bendell et al., 2012*; *Hong et al., 2012*; *Juric et al., 2015*). Since our submission, the FDA-approved alpelisib for pediatric and adult patients with PROS, and several clinical trials are underway to assess safety, efficacy, and quality-of-life with alpelisib in patients with a PROS diagnosis (e.g., NCT04589650, NCT04085653, NCT04980833, and NCT05294289), demonstrating the swiftness of efforts to address this clinical need.

Furthermore, in our WGS analysis, we identified only a few somatic variants within protein-coding genetic sequences (*Supplementary file 1*) beyond what was reported in the cancer gene panel (*Table 1*). The low frequency of somatic mutations is consistent with findings in other low-grade pediatric tumors (*Akhavanfard et al., 2020*). In addition to detecting the *PIK3CA* H1047R mutation, this WGS confirmed the variant detected by the cancer gene panel in the *PIK3C2B* gene and demonstrated that it was germline. Although this in *PIK3C2B* variant has not been characterized and may be a benign polymorphism, this finding raises the issue of whether other alterations in the pathway may cooperate with activating mutations of *PIK3CA* to induce cell proliferation. The low VAF driver mutations in tissue derived from LMs is likely due to the fact that most pathological tissue is composed of reactive stromal elements while the clonal cells represent a relatively small portion (presumably the lymphatic channel-lining endothelial cells). Consistent with this observation, in the alpelisib-treated index patient, we observed most intense activation of the PI3Kα pathway in these lymphatic channel-lining endothelial cells (*Figure 2F, G*). The high representations of pathways associated with vascular development, cell motility, inflammatory response, positive regulation of response to stimuli, blood vessel morphogenesis in our gene expression analysis are consistent with a mechanistic hypothesis that most of the lesion represents an intense reactive response to the (presumably) clonal LM-LECs, although the appropriate comparator control tissues for these lesions is not clear.

Evidence is accumulating that a variety of 'nonmalignant' syndromes associated with abnormal tissue growth may be driven by underlying alterations in classic oncogenes (*Mustjoki and Young, 2021*). *PIK3CA* mutations are seen not only in LMs but other vascular anomalies, highlighting the role of *PIK3CA* activation in angiogenesis, lymphangiogenesis, and vascular neoplasms (*Castel et al., 2016*; *Castillo et al., 2016*; *Limaye et al., 2015*; *Ren et al., 2021*). Endometriosis, uterine fibroids, and seborrheic keratoses all have been found to harbor mutations in cancer-related genes (*Rafnar et al., 2018*; *Gallagher et al., 2019*; *Fritsche et al., 2018*; *Sanders et al., 2018*; *Anglesio et al., 2017*). These findings suggest that targeted therapies being developed for invasive cancers may also be active in proliferative lesions that are not classified as invasive cancers that harbor the targeted alteration.

In summary (*Figure 5*), we find that the majority of LMs have driver mutations that are potentially targetable. LMs with classic histology mostly have *PIK3CA* mutations that may respond to alpelisib. LMs with kaposiform histopathology are enriched in *NRAS* mutations, and studies are required to determine if these may respond to clinically available MEK inhibitors. LMs that are wild-type for

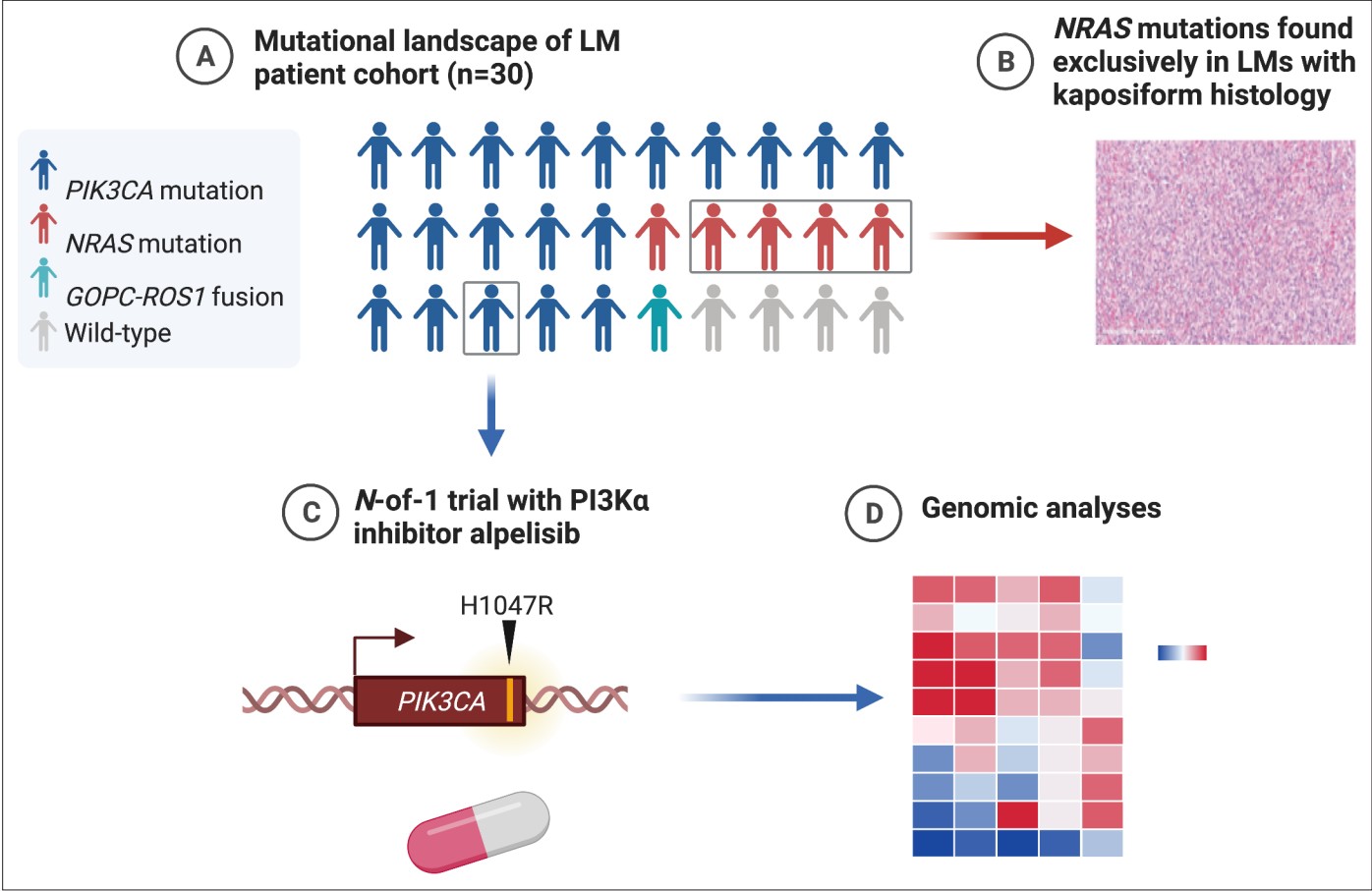

**Figure 5.** Graphical summary of the mutations found in genomic analysis of lymphatic malformation (LM) patient cohort (created with BioRender.com). (**A**) The majority of LMs have driver mutations that are potentially targetable. (**B**) LMs with *NRAS* mutations had kaposiform histopathology. (**C**) An *N*-of-1 clinical trial is reported in a patient with a targetable PIK3CA mutation. (**D**) Comprehensive genomic analyses may reveal further actionable molecular insights.

*PIK3CA* and *NRAS* may have other actionable alterations, such as the *GOPC–ROS1* fusion seen in our series and may require more comprehensive genomic analyses to identify them. Systemic treatment with targeted therapy aimed at the driver mutation in LMs may be an option for some patients who are not controlled by surgery and other conventional treatments.

## Acknowledgements
We thank Dr. Kathryn J Brayer for technical assistance and Dr. Helen Pickersgill (Life Science Editors) for professional manuscript editing services.

## Additional information

### Competing interests
Montaser F Shaheen: reports personal fees from Illumina, BMS, and Qiagen (outside of the submitted work). The author has no other competing interests to declare. Julie Y Tse, Ethan S Sokol: is an employee of Foundation Medicine, Inc, a wholly owned subsidiary of Roche, and owns equity in Roche. The author has no other competing interests to declare. Brian K Mannakee: is an employee of Foundation Medicine, Inc, a wholly owned subsidiary of Roche, and owns equity in Roche. Roman Groisberg: reports research funding/grant support for clinical trials (to his institution) from Regeneron, BMS, Merck/EMD Serano, Amgen, Roche/Genentech, Philogen; consulting/advisory board

fees from Regeneron; and speaker fees for Deciphera (all outside of the submitted work). These arrangements are managed in accordance with the established institutional conflict of interest policies of Rutgers, The State University of New Jersey. The author has no other competing interests to declare. Renata Pasqualini, Wadih Arap: Reviewing editor, *eLife*. Shridar Ganesan: has consulting agreements with Merck, Roche, Novartis, Foundation Medicine, EQRX, Foghorn Therapeutics, Silagene, and KayoThera and owns equity in Silagene; his spouse is an employee of Merck and owns equity in Merck (all outside of the submitted work). These arrangements are managed in accordance with the established institutional conflict of interest policies of Rutgers, The State University of New Jersey. The author has no other competing interests to declare. The other authors declare that no competing interests exist.

## Funding

| Funder | Grant reference number | Author |
|---|---|---|
| Levy-Longenbaugh Fund | | Renata Pasqualini Wadih Arap |
| Hugs for Brady Foundation | | Shridar Ganesan |
| National Cancer Institute | P30CA072720 | Renata Pasqualini Shridar Ganesan Wadih Arap |
| National Science Foundation | DGE-1143953 | Brian K Mannakee |
| National Cancer Institute | P30CA023074 | Montaser F Shaheen |
| National Cancer Institute | P30CA118100 | Scott A Ness |

The funders had no role in study design, data collection, and interpretation, or the decision to submit the work for publication.

## Author contributions

Montaser F Shaheen, Conceptualization, Funding acquisition, Investigation, Methodology, Project administration, Supervision, Visualization, Writing – original draft, Writing – review and editing; Julie Y Tse, Conceptualization, Investigation, Methodology, Project administration, Supervision, Visualization, Writing – original draft, Writing – review and editing; Ethan S Sokol, Conceptualization, Investigation, Methodology, Visualization, Writing – original draft, Writing – review and editing; Margaret Masterson, Conceptualization, Funding acquisition, Writing – review and editing; Pranshu Bansal, Ian Rabinowitz, Christy A Tarleton, Andrey S Dobroff, Tracey L Smith, Thèrése J Bocklage, Brian K Mannakee, Ryan N Gutenkunst, Joyce Bischoff, Scott A Ness, Gregory M Riedlinger, Roman Groisberg, Investigation, Methodology, Visualization, Writing – review and editing; Renata Pasqualini, Shridar Ganesan, Wadih Arap, Conceptualization, Funding acquisition, Project administration, Supervision, Writing – original draft, Writing – review and editing

## Author ORCIDs

Andrey S Dobroff (iD) http://orcid.org/0000-0003-2162-9951
Ryan N Gutenkunst (iD) http://orcid.org/0000-0002-8659-0579
Joyce Bischoff (iD) http://orcid.org/0000-0002-6367-1974
Wadih Arap (iD) http://orcid.org/0000-0002-8686-4584

## Ethics

Clinical trial registration NCT03941782.

Approval for this study, including a waiver of informed consent and Health Insurance Portability and Accountability Act waiver of authorization, was obtained from the Western Institutional Review Board (IRB; protocol #20152817). A single-institution personalized clinical protocol to treat the patient with the experimental PI3Ka inhibitor alpelisib was scientifically reviewed by the Protocol Review and Monitoring Committee (PRMC) and approved by the local Institutional Review Board (IRB) of the University of New Mexico Comprehensive Cancer Center. The study (NCT03941782) was conducted in accordance with the protocol, Good Clinical Practice guidelines, and the provisions of the Declaration of Helsinki. The index patient signed an informed written consent form.

Decision letter and Author response
Decision letter https://doi.org/10.7554/eLife.74510.sa1
Author response https://doi.org/10.7554/eLife.74510.sa2

## Additional files

### Supplementary files
• Supplementary file 1. Somatic coding mutations identified from whole-genome sequencing. The genetic coding variants that exist in lymphatic malformation (LM) but do not exist in germline DNA. These pass MuTect2 quality filters (designed to call somatic variants only) and have three or more alternate reads. VAF, variant allele frequency; COSMIC, Catalogue Of Somatic Mutations In Cancer.
• Transparent reporting form

### Data availability
NGS data for this study were generated at Foundation Medicine, Inc on U.S. patients profiled during routine clinical care. Approval for this study, including a waiver of informed consent and Health Insurance Portability and Accountability Act waiver of authorization, was obtained from the Western Institutional Review Board (protocol #20152817). Due to potential for identifiability, patient-level alteration data are not available. However, extensive data supporting the findings of this study are available in Table 1 and Figure 1A.

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
