## [Editor Report]

The study examines the genomic landscape of a patient cohort of lymphatic malformations (LMs) through next-generation sequencing and immunocytochemistry. The authors identified actionable driver mutations in the P13KCA and NRAS genes. The study enhances our understanding of the genetic architecture of the otherwise disfiguring LMs in people.

---

## [Decision Letter]

**Decision letter after peer review:**

Thank you for submitting your article "Genomic Landscape of Lymphatic Malformations: A Case Series and Response to the PI3Kα Inhibitor Alpelisib in an N-of-One Clinical Trial" for consideration by *eLife*. Your article has been reviewed by 2 peer reviewers, and the evaluation has been overseen by a Reviewing Editor and Martin Pollak as the Senior Editor. The following individual involved in the review of your submission has agreed to reveal their identity: Friedrich Kapp (Reviewer #1).

Below, I have provided a summary of essential revisions. Please provide a point-by-point rebuttal, also taking into account comments made under the heading 'Public Review', as appropriate changes in response would strengthen the manuscript further.

Essential revisions:

1. Please provide a more accurate description and naming of vascular malformations, as well as distinguish these from tumors.

2. The abstract would be improved if it included more specific aspects of the results, e.g. the number of patients with PIK3CA and NRAS mutations and what they mean by "complete response" and "sensitive" to alpelisib.

3. The CT scan images should indicate the LM with arrows.

---

## [Author Response]

Essential revisions:1. Please provide a more accurate description and naming of vascular malformations, as well as distinguish these from tumors.

Thank you for the suggestion to clarify the nomenclature used in this manuscript for the non-expert reader. We considered the diagnosis and classification of vascular anomalies (vascular malformations and others) to be a holistic integration of clinical examination, imaging studies, pathology diagnosis, and/or genomic results. One of the limitations of our study is that the CGP cohort was a study of data available from an international reference laboratory. While the use of data from a reference laboratory enables the study of relatively high numbers of rare diseases, it limits us to only clinical information provided by the ordering physicians at the time of testing and only to one representative pathology specimen submitted by the pathology laboratory. Therefore, the scope of our study did not enable outreach to ordering physicians and pathologists to determine if and how the genomic results refined the working clinical diagnosis and/or pathologic diagnoses.

Recognizing these inherent limitations, we agreed to use the term lymphatic malformations to describe the lesions in our cohort. Lymphatic malformation is widely accepted to include a clinicopathologic continuum of benign tumors of lymphatic origin (https://rarediseases.org/rare-diseases/lymphatic-malformations/), including cystic lymphangioma, kaposiform lymphangiomatosis, macro/microcystic lymphatic malformation. While evidently imperfect, we have now clarified their use of this term in the Introduction.

2. The abstract would be improved if it included more specific aspects of the results, e.g. the number of patients with PIK3CA and NRAS mutations and what they mean by "complete response" and "sensitive" to alpelisib.

Thank you for the suggestion. We have specified the number of patients in the CGP cohort with PIK3CA and NRAS mutations in the Abstract. We have also clarified the language “complete response” and “sensitive” to treatment.

3. The CT scan images should indicate the LM with arrows.

We have now indicated the LM in the CT images to allow easier visualization for the readers.